# A long record of European windstorm losses, and its comparison to standard climate indices

Stephen Cusack

Stormwise Ltd, Luton, LU4 9DU, United Kingdom

*Correspondence to*: Stephen Cusack (stephen.cusack@stormwise.co.uk)

**Abstract.**

Traditional insurance has both a great exposure to decadal variations in European storm activity, and the ability to adjust its business strategy over these timescales. Hence, the recent development of skilful predictions of multiannual mean European winter climate seems a very welcome addition to the long list of ways that researchers have improved management of

windstorm risk. Yet companies do not use these forecasts of mean winter climate to adjust their view of risk. The main reason is the lack of a long, reliable record of losses to understand how forecasted time-mean circulation anomalies relate to the damage from a few, intense storms. This study fills that gap with a European windstorm loss record from 1950 to 2022, based on ERA5 peak near-surface winds per event which were converted to losses using an established damage function. The resulting dataset successfully identifies major storms over the past 70 years, and simulates the multidecadal variations from

low values in the 1960s up to high levels in the 1980s and '90s then down to the 2010s. However, it underestimated the steepness of the observed loss decline from the stormy end of the 20th century to the lull over the past 20 years. This was caused by a quite flat trend in ERA5 extreme winds over the period, in contrast to the significant decline in observed peak gusts. Imposing these gust trends on ERA5 peak winds reconciled modelled losses with industry experience over the past few decades.

Indices of European winter climate used in long-range forecasting were compared to the new modelled loss dataset. They had correlations of around 0.4 at interannual timescales, rising to about 0.7 for decadal and longer variations. Notably, the climate indices have a similar multidecadal trend as ERA5 extreme winds in modern times, including a less steep decline than found in observed gusts and losses. Further investigation of the modern-day divergence between climate indices and losses may help connect decadal forecasting to insurance.

## 1 Introduction

Extreme wind gusts from winter storms cause much damage to Europe. For example, Barredo (2010) examined Europe-wide economic losses in the Munich Re NATHAN catastrophe database for the period 1970-2008 and, when trended to 2022 using 5% p.a. growth (e.g. Klawa and Ulbrich, 2003), there are four storms with losses exceeding 20 billion USD (Capella in 1976,

87J in 1987, Daria in 1990, and Lothar in 1999), and the annual average from the top 25 events is around 7 billion USD. Such large impacts stimulate research into Europe's climate of extreme windstorms.

The property insurance sector covers around one half of these losses (e.g. Guha-Sapir et al., 2022), and therefore has keen interest in this risk. The majority of the market is classed as traditional insurance, while the more modern insurance-linked securities (ILS) segment provides additional capacity from outside investors. Traditional insurance manages their exposure according to a view of the windstorm risk typically over the next five or so years, corresponding to their review cycle of weather perils, whereas ILS is more active at annual timescales with a significant fraction tradeable much more frequently.

There is a long history of collaboration between researchers and the insurance sector towards a better understanding of this peril. For example, academics eased industry concerns by placing two violent British storms in a longer-term context (Christofides et al., 1992), and three decades later, knowledge of past storms continues to grow (e.g. Hawkins et al., 2023). Multiple occurrences of damaging storms stressed the sector in 1990 and 1999, and led to greater insights into the clustering phenomenon (e.g. Mailier et al., 2006; Karremann et al. 2014; Cusack, 2016; Priestley et al., 2018). More recently, insurance has been considering how to manage the risk posed by anthropogenic climate change (e.g. EIOPA, 2021) and studies of mid latitude windstorm impacts have proven valuable (e.g. Chang, 2018; Bueler and Pfahl, 2019; Catto et al., 2019; Harvey et al., 2020). Research has delivered many actionable insights for insurance.

Over the past decade or so, there has been a growing awareness of a multidecadal variability of storm activity over Europe which is both material to risk, and slow enough for insurance companies to adapt to it. The evidence of such behaviour is found in a wide variety of studies (e.g. WASA Group, 1998, Dawson et al., 2002, Brázdil et al., 2004; Cusack, 2013; Stucki et al., 2014; Feser et al., 2015; Dawkins et al., 2016; Laurila et al., 2021). More specifically to insurance, Cusack (2013) focused on a measure reflecting insured losses and found the decadal-mean windstorm damage in the Netherlands contained variations with amplitudes exceeding a factor two. The existence of slow variations in storm activity, over the past 100 and more years, is firmly established by many studies.

In parallel, there has been much progress in identifying and understanding drivers of these decadal storm variations, as recently reviewed by Cassou et al. (2018). Researchers have found decadal and longer timescale anomalies being driven by North Atlantic Ocean heat contents (Omrani et al., 2014; Peings and Magnusdottir, 2014; Hu et al., 2019), Arctic sea ice extent (e.g. Smith et al., 2022) and thickness (e.g. Lang et al., 2017) or more generally Arctic change (e.g. Cohen et al., 2018), and their interaction via ocean heat transport through the Norwegian Sea into the Arctic Seas (Zhang, 2015; Årthun et al., 2017). Other parts of the climate system have also been associated with decadal and longer changes in mid-latitude storm activity, such as the tropics (Greatbatch et al., 2012), the stratosphere (Scaife et al., 2005), anthropogenic gases (e.g. Shaw et al., 2016), and major volcanoes (Swingedouw et al., 2015).

The documented decadal variability and process-based research are now complemented by skilful forecasting of the dominant mode of winter-mean variability in the North Atlantic sector, the North Atlantic Oscillation (NAO). Decadal prediction systems integrate the set of diverse drivers into a view of the future climate, and skilful North Atlantic decadal forecasts were first reported in Keenleyside et al. (2008). A study by Eade et al. (2014) found prediction systems contained more skill than implied

by rather low signal-to-noise ratios, and suggested large ensembles of forecasts could realise this potential. Recently, hindcast tests conducted by Athanasiadis et al. (2020) found correlations up to 0.63 between forecasted and observed decadal NAO anomalies for a 40-member ensemble, and promising signs of higher correlations from larger ensembles. Indeed, Smith et al. (2020) found correlations of 0.79 based on an ensemble with 676 members.

Despite these advances in characterising and understanding decadal climate variability in Europe and the wider North Atlantic sector, and skilful decadal forecasting, there has been no known application of real-time predictions to European windstorm insurance. The main reason is the uncertain relation between predicted indices – essentially large scale, winter-mean circulation anomalies – used in forecasting, and the windstorm losses of concern to insurance and caused by a few intense weather events. More generally, published research into storminess uses a much broader array of metrics, in addition to the NAO mentioned above. For example, the basic observed quantity may range from meteorological variables such as wind speeds at the near-surface (Smits et al. 2005, Chang, 2018) or upper troposphere (Harvey et al. 2020), vorticity at 850 hPa (Deroche et al. 2014) and geostrophic winds derived from surface pressure gradients (WASA Group 1998), or they may be based on economic losses (Barredo, 2010), or forestry damage (Gregow et al. 2017). Quantities are processed in a variety of ways to produce a measure of windiness. While all metrics have a reasonable relation to storminess, their precise relations to losses are unknown. Insurance companies are cautious about using diagnostics with unknown relation to loss because they may suffer severe penalties for mis-priced risk.

This barrier between research and applications would be removed with a long windstorm loss history to assess the storm metrics. However, publicly available information on losses is limited. For example, the EM-DAT database (Guha-Sapir et al., 2022) contains reliable information for some major storms, but misses national losses for many significant events in the past 50 years. PERILS (available from https://www.perils.org/losses) provides reliable estimates of insured losses covering the countries contributing the vast majority of Europe-wide insured losses, for significant storms since 2009, and five earlier storms in the previous ten years. It is encouraging for the future, but too short at present for robust assessment of storm metrics.

The main aim of this study is to develop a comprehensive windstorm loss history for Europe in the period 1950-2022, using near-surface winds from reanalyses in combination with a standard method to convert windspeeds to losses. Section 2 provides details on the data used in this study, while Section 3 describes how storm losses are computed. Section 4 contains an initial evaluation of its losses with insurance industry knowledge, together with further development to reproduce multidecadal trends in observed losses. A comparison of commonly used climate indices to these calibrated losses is given in Section 5, then the main conclusions are presented in Section 6.

## 2 Datasets and processing

Windstorm damages are computed over the domain of 12 countries shown in Figure 1, including those that are key to European windstorm insurance, such as Germany, France, United Kingdom and Netherlands. Other countries will experience smaller

insured losses due to either their weaker wind climate, or smaller population, or reduced insurance penetration. The industry

considers the highlighted region in Figure 1 to incur the vast majority of European windstorm *insured* losses.

## 2.1 Loss data

Observed winterstorm losses are developed from a published loss in a given year, then indexed to the common year of 2022 to remove the effects of societal changes on loss estimates. All loss data in this study are indexed from 1999 onwards, therefore

trends from that year to 2022 need defined. Klawa and Ulbrich (2003) chose 5% p.a. trend for the period 1970-1999, a value that was positioned between inflation rates for buildings in Germany, and sharper increases of 7% p.a. through the 1990s reported by Munich Re (2002). Trends in the 21$^{st}$ century are expected to be lower for two main reasons: (i) inflation was generally higher in the 1970s through to the early 1990s than from 2000 to 2022, (ii) Munich Re trends include growth in exposure from increased uptake of insurance, and re-unification of Germany, both of which do not contribute to growth over

the past two decades.

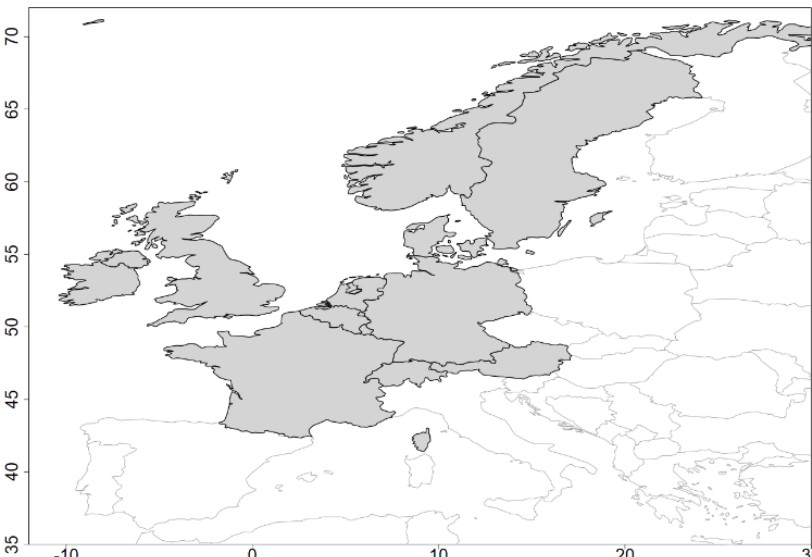

**Figure 1: a map of Europe, with the shaded area showing the 12 countries studied here.**

An appropriate trend for the 21$^{st}$ century is developed by decomposing total losses into trends in frequency and severity of claims. Both general measures of inflation and slightly more relevant construction cost indices suggest increases in claim severity of 2 to 3% p.a. from 2000 to 2022. The growth in claim frequency is expected to be more muted than in the 20$^{th}$ century due to slower population growth and near-saturation of windstorm insurance uptake in these markets. Instead, the growth in number of claims in the 21$^{st}$ century is more driven by the growth in the number of properties. Data on the total

number of dwellings in Germany (from https://www.destatis.de/EN/Themes/Society-Environment/Housing/_node.html) and the U.K. (e.g. https://www.gov.uk/government/statistical-data-sets/live-tables-on-dwelling-stock-including-vacants) indicate 0.5 to 1% p.a. growth over the past 20 or more years. In this study, the claims frequency and severity components are combined into a uniform trend of 3.5% p.a. from 1999 to 2022.

For comparison, the work by Pielke and Landsea (1998) considered three factors, namely inflation, population, and wealth.

The combination of the three factors produces a quantity akin to the nominal Gross Domestic Product (GDP), and the explicit partitioning in Pielke and Landsea (1998) served to distinguish their analysis from previous published work which largely relied on inflation alone. An analysis of GDP figures from the OECD (available from https://data.oecd.org/gdp/gross-domestic-product-gdp.htm) for the three biggest countries (France, Germany, U.K.) reveals a range of 4.1 to 4.3% p.a. growth in GDP over the period 2000 to 2022. However, the GDP includes a measure of growth in wealth/possessions (aka real GDP per capita)

which is of little relevance to most windstorm claims mainly consisting of tile damage. Data from Eurostat (https://ec.europa.eu/eurostat/databrowser/view/sdg_08_10/default/table?lang=en) indicate the real GDP per capita has been growing at an average rate of just under 1% p.a. in the $21^{st}$ century, hence the lower growth rate of 3.5% p.a. used in this study is more consistent with the factors driving trends in windstorm damages. Other types of catastrophes causing more severe damage to properties, such as flood or tropical cyclones, would be more appropriately trended using nominal GDP.

The set of reported losses can be split into two groups: more modern storms in 2009-2022 taken from PERILS (available from https://www.perils.org/losses) and older storms which have been intensively studied due to their severe impacts. Losses for both groups are based on market surveys following the event, and due to strict financial standards, these *reported* losses are likely to be thoroughly reviewed internally, and accurate. However, the trending of the losses to a common year is a significant source of uncertainty, more so for older storms. For example, the true average $21^{st}$ century trend in total losses could be as low

as 2.5% or as high as 4.5% p.a. based on claims frequency and severity data discussed above, and there is additional uncertainty since most damage concerns roof tile replacement and much narrower than broader inflation or construction cost measures. These considerations suggest the potential bias in indexed losses is reasonably approximated as growing by 1% per year. Therefore, more modern PERILS loss estimates could be biased by around 10%, while damage estimates of older storms are expected to be more uncertain.

Table 1 summarises the loss data used to evaluate the new dataset. PERILS losses represent the same domain highlighted in Figure 1, and include those events causing more than €200 million insured loss. The same loss threshold of €200 million will be applied to events in the new dataset for the loss validation performed in Section 4. All other sources of loss estimates may potentially include losses outside of the 12 countries, though such an additional contribution is expected to be minor and within uncertainty, especially for older storms. Multiple sources are available from Swiss Re over the years, and the earliest values

are chosen so as to minimise the use of their trending based on the United States consumer price index (Swiss Re, 2002).

**Table 1: historical loss data used for validation.**

| Storms | Loss estimate in € billions, and source | Final estimate in € billions, and range |
|---|---|---|
| 2009-2022 | PERILS reported, + 3.5% p.a. to 2022 | PERILS (± 10%) |
| Kyrill, Jan 2007 | 6.12 (PERILS)<br>6.95 (Swiss Re, 2008) | 6.5 (± 15%) |
| Dec 1999 | 22.2 (PERILS)<br>23.5 (Munich Re, 2002)<br>22.7 (Swiss Re, 2002)<br>22.2 (RMS[a]) | 22 (± 20%) |
| Jan-Mar 1990 | 35.3 (Munich Re, 2002)<br>27.3 (Roberts et al., 2014; extracted from Swiss Re)<br>34.7 (RMS, 2007[b]) | 30 (± 20%) |
| 87J | 9.0 (Munich Re, 2002)<br>8.25 (Swiss Re, 2002)<br>9.6 (RMS, 2007) | 9 (± 20%) |
| Capella | 9.0 (Munich Re, 2002) | 9 (± 30%) |

[a] https://www.rms.com/blog/2019/12/18/twenty-years-after-storms-anatol-lothar-and-martin-memories-from-the-end-of-the-millennium

[b] RMS estimate of storm Daria inflated to total 1990 loss using its fractional contribution to the total from Munich Re (2002)

## 2.2 Wind data

The windstorm losses from 1950 to 2022 are based on instantaneous (12-minute timestep) winds at 10m above short grass,
available from ERA5 (Bell et al., 2021) at hourly frequency. Peak gusts are more commonly associated with damage (e.g. Prahl et al., 2015) however, it was found that storms based on ERA5 peak gusts (ECMWF, 2016) validated poorly. For example, footprints based on ERA5 peak gusts indicate storm Kyrill produced the biggest loss in the past 70 years, and 72% higher than those of Daria (1990), while storm 87J (1987) was a lowly rank 42 and just 12% of Kyrill's loss. In sharp contrast, observed losses (Table 1) indicate storm 87J and Capella had greater losses than Kyrill, and breakdowns by event for the 1990
and 1999 seasons from both Munich Re and Swiss Re indicate Daria and Lothar losses are higher again. As a result of intensive research into these extreme storms, there is a high level of confidence that the true storm loss relativities are very different from those based on ERA5 gusts. Losses based on ERA5 near-surface winds will be shown later to be more consistent with experience than those based on ERA5 gusts.

Observations in the Integrated Surface Database (ISD, Smith et al., 2011) from the National Oceanic and Atmospheric
Administration (NOAA) National Centre for Environmental Information (NCEI) are also analysed. The ISD is a global collection of surface-based observations built from many national sources, with its specific purpose to homogenise the mixture of data formats used worldwide, and thereby ease access for researchers. It contains observations at hourly to 6-hourly frequency from over 20,000 weather stations which have been active for some time within the period from 1900 to the present day. ISD offer additional products in line with their aim to make global surface weather observations more accessible. The

Global Summary of the Day (GSOD) is a more concise version of ISD with fewer variables at daily resolution, and the daily maximum values of peak gusts in this dataset are used later in this study.

### 2.3 Climate index data

The three most commonly used indices of mean circulation anomalies over Europe, and often linked to regional storminess,
are the NAO, Scandinavian Pattern (SCA) and the Arctic Oscillation (AO). NAO data (Hurrell et al., 2003) are provided by NCAR (available at https://climatedataguide.ucar.edu/climate-data/) and both the principal component (PC) and station-based (Lisbon and Iceland) versions are assessed. Monthly mean pressures at sea level have been extracted from ERA5 archives to define the two other indices. The SCA values (Barnston and Livezey, 1987) are computed as the difference between a southern (35° N to 50° N, 10° E to 30° E) and northern box (60° N to 75° N, 10° E to 30° E), while AO values are computed as the
average value over the North Pole (north of 60°N) of mean sea level pressure from ERA5.

### 2.4 Data processing

All monthly data are processed into storm seasons by averaging monthly values from October to March, except for the storm index which includes those few extreme events occurring in April.
The main focus of this study are those variations at decadal and longer timescales which can be assimilated into the ca. 5-year pricing review cycle of most insurance business. These variations are examined using low-pass filtered versions of annual timeseries, produced by applying a fourth-order Butterworth filter with a 10-year cutoff frequency.
Some of the later analysis converts timeseries to standardised anomalies (mean of zero, standard deviation of one) using their sample statistics, to enable comparison of quantities with different units, such as storm losses and climate indices.

### 3 Defining storm event losses

The maximum values of ERA5 near-surface winds were computed at daily resolution for every grid cell in the domain, and a proxy of damage (D) for the entire domain was defined for every day from 1950 to 2022 as follows:

$$D = \sum_{i=1}^{N} \left[ max \left( \frac{v_{i,d}}{v_{i,98}} - 1, 0 \right) \right]^3$$

where there are N grid cells in the domain, $v_{i,d}$ is the daily maximum wind, and $v_{i,98}$ is the climatological 98th percentile of wind. This daily damage quantity is based on the loss proxy discussed below.

The next step is to form storm events as a series of up to three days centred on the days with peak values of D, then compute the event maximum wind ($v_{i,s}$) over the days of the storm, for each grid cell. The outcome is a matrix of peak windspeeds stored for each grid cell, and every storm.

The grid cell peak storm winds were converted to domain-wide event losses using the model from Klawa and Ulbrich (2003):

$$L_s = c. \sum_{i=1}^{N} P_i \left[ max \left( \frac{v_{i,s}}{v_{i,98}} - 1, 0 \right) \right]^3$$

where $L_s$ is the loss for storm s, $P_i$ is the population count for the i'th cell, and c is a constant of proportionality intended to re-scale values to represent losses. The population data is from Gridded Population of the World, version 4, at 2.5 minutes of arc resolution (CIESIN, 2018). The constant of proportionality c in this model is used to scale values to losses, and in this study was defined to reproduce a €30 billion aggregate wind loss in the 1989/90 season, from Table 1. Finally, event losses in the October to April period were summed together to form total damage per windstorm season.

The above equation defines losses to vary with the cube of the wind excess above the local 98th percentile, and population density. Klawa and Ulbrich (2003) discussed the basis of this formulation, together with a validation based on 20 years of industry-wide insured losses in Germany, and other selected events. Another feature of this loss equation, of more relevance later, is the assumption that the size of loss for a given wind speed is independent of time. Whether this assumption holds over the 1950 to 2022 period is unclear, and discussed in greater depth in the next section.

## 4 A European windstorm loss history

### 4.1 Initial evaluation of historical losses

Figure 2 shows losses per windstorm season from 1950/51 to 2021/22 using the ERA5 winds. At a high level, the model compares well to observed losses for key storms: the historic years of 1990 and 1999 are prominent, as are other large loss years containing landmark storms (most notably the Lower Saxony storm in November 1972; Capella in January 1976, and the stormy period in the early 1980s following the El Chichón eruption; and Kyrill in January 2007).

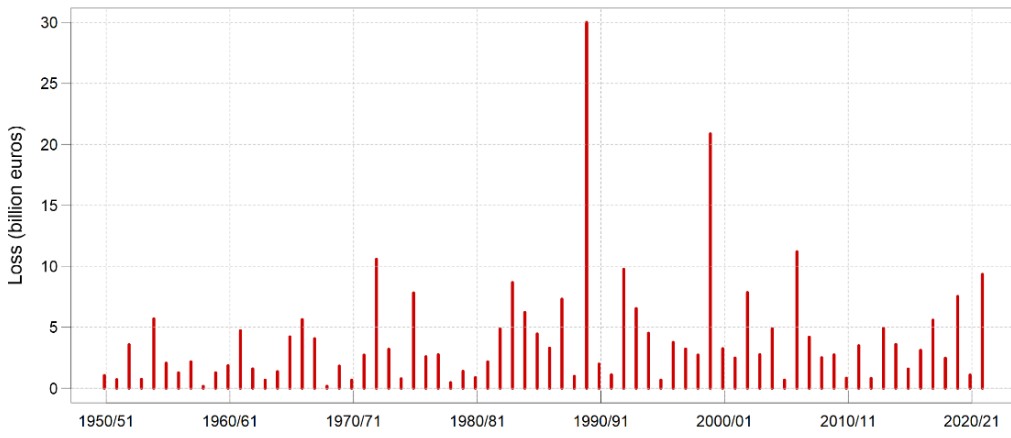

**Figure 2: barplot of Europe-wide annual windstorm losses, from 1950/51 to 2021/22.**


More detailed loss validation was performed by comparing with PERILS insured losses in Figure 3. The new dataset of losses exceeds those of PERILS in all but one year, and the probability of 13 or more years being higher out of 14 years, when both are drawn from the same parent distribution, is 0.00092 from the binomial distribution. There is a corresponding high bias of 107% over the 14-year period, and their means are significantly different at the 5% level. The evidence that modelled losses

exceed observed in 2009-2022 is compelling. On the other hand, the modelled losses for older storms do not have such a high bias versus observed values: both the €30 billion for 1990 and a little over €20 billion for 1999 are consistent with observed (the former by design), while modelled losses for Capella in 1976 and 87J in 1987 are more than 20% **lower** than the observed values in Table 1.

Additional evidence on this contrast between older and newer storms can be gained from Figure A2 of Barredo (2010). While

Barredo's absolute values are not suitable because they concern economic rather than insured losses, they can inform on the relativity between storms, and suggests losses in 1976 were double those in 2007, and losses in 1987 exceeded those in 2007. The values in Figure 2 place 2006/07 losses around 30% above those two earlier seasons, and not consistent with values in Barredo (2010), as well as Table 1.

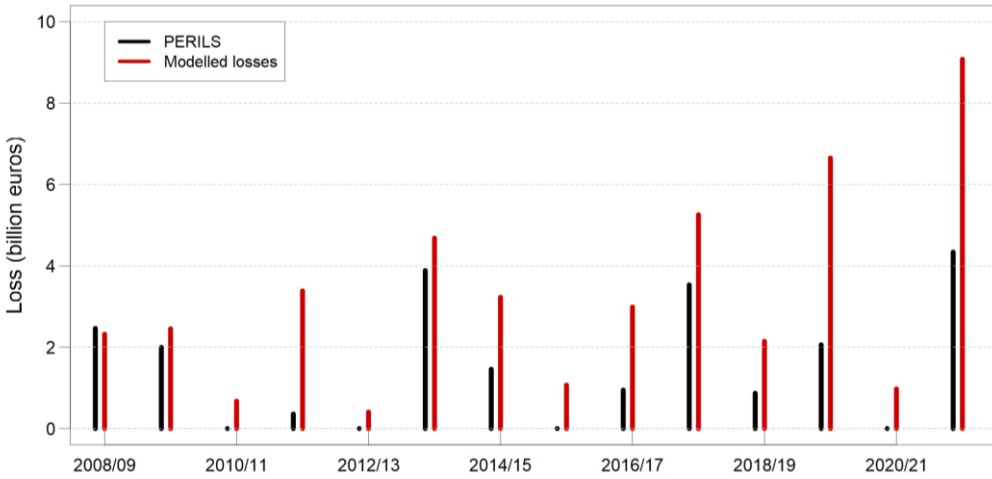


**Figure 3: a barplot of Europe-wide windstorm losses for the past 14 seasons, from PERILS and the new dataset.**

The evidence clearly indicates modelled losses based on ERA5 winds tend to be low for earlier landmark storms, then evolve into more than double those observed in the most recent period. The growing positive bias in modelled losses in recent times

is substantial, and is investigated further in the next subsection.

### 4.2 Multidecadal trends in losses

The mis-matched values of the loss decline from the 1980s and '90s to the 2010s could be caused by one or more contributions from three different drivers: (i) the observed losses used in validation contain too great a decline from the last two decades of the 20[th] century to the 2010s, or (ii) the relation between hazard and loss is non-stationary, whereas the modelling presumes it

is homogeneous, or (iii) ERA5 winds are not representing the true trend in the hazard quantity causing windstorm damage. These three possibilities are now investigated.

There is high confidence in the observed data which define the declining trend in losses, and as a result the first potential driver above is considered to contribute little to different trends in observed and modelled. Both older and modern observed losses are based on surveys of companies incurring losses, and both regulatory controls and intense scrutiny of the older, extreme

events ensure the industry has accurate information on reported losses. It is also considered unlikely that an overestimation of trending of older costs to the present day is responsible. For example, the reported losses in 1990 from Munich Re (2002) would need trended by 1% p.a. to produce a relativity to the 2021/22 losses similar to the modelled value (a factor of three). Such low indexation does not fit with observed data: for instance, the factor 1.8 to 2 in the decade of the 1990s found by Munich Re (2002) suggests a 1% p.a. trend is not credible.

The second potential driver would be wholly responsible if the hazard produced twice as much loss in the 1970s to 1990s period, than if the same hazard occurred in the present day. Possible causes of such a non-stationarity include changes in building practices which alter the vulnerability of roofs to damage from winds, or a social change in claiming practice (such

as the insured's disposition towards making a claim) or repair methods (e.g. modified safety regulations governing roof repair). Such non-meteorological drivers of severe weather loss trends have been detected in the United States (e.g. RMS, at

https://www.rms.com/blog/2018/08/03/us-severe-convective-storm-claims-going-through-the-roof) though they act to boost rather than reduce modern day losses, hence the opposite direction from what is required to explain the deficit in the modelled loss trend. Irrespective, there is no empirical evidence of this type of loss inhomogeneity for Europe windstorms. On the contrary, there is indirect evidence of a stationary relation from hazard to loss from catastrophe modelling vendors. For example, the RMS estimates of storm losses in Table 1 are based upon reported hazard, which are input into damage functions calibrated to losses chiefly from modern times, and this produces total losses which fit with other empirical estimates (see Table 1). Therefore, the existing evidence suggests this driver contributes little to the tendency for the modelled dataset to have a different loss trend from observed.

The above considerations led to a deeper study of the third potential driver: whether ERA5 winds could contribute to an underestimated decline in losses. There are two different types of evidence suggesting winds from reanalyses may provide imperfect long-term trends. First, it is well established how reanalyses may contain non-meteorological trends due to changing observation systems, depending on meteorological variable, time period and region being studied (e.g. Bengtsson et al., 2004; Thorne and Vose, 2010). Second, storm damage is most closely associated with gusts rather than winds, and there is potential for these two quantities to have different trends, since gusts depend on additional small-scale processes which are less well resolved by mean winds representing longer timescales. Given these uncertainties in reanalyses and time-mean winds, multidecadal trends in ERA5 winds have been compared to those from observed gusts.

Figure 4a shows the trend in ISD gusts from 1980 to 2022 for those 221 stations with at least 38 years of non-missing gust data in this time period. Annual averages of the top five gusts per year were computed, then re-scaled with the station's long-term mean value of this quantity, so that the resulting fitted linear trends at each station can be expressed in units of % change per year. The method was repeated for ERA5 reanalyses winds at the same locations as the surface weather stations, and their trends are shown in Figure 4b.

In general, the plots show a magnitude of downward trend in observed extreme gusts which is not replicated in ERA5 reanalyses winds. For example, observed gusts have a more negative trend than ERA5 winds at 180 of the 221 stations (Figure 4c), and such a preponderance of more negative gust trends has a vanishingly small chance of being produced by random sampling error. The U.K. data are now selected to study this issue in more detail, because of its good station density (51 in total) and relatively consistent signal in Figure 4c.

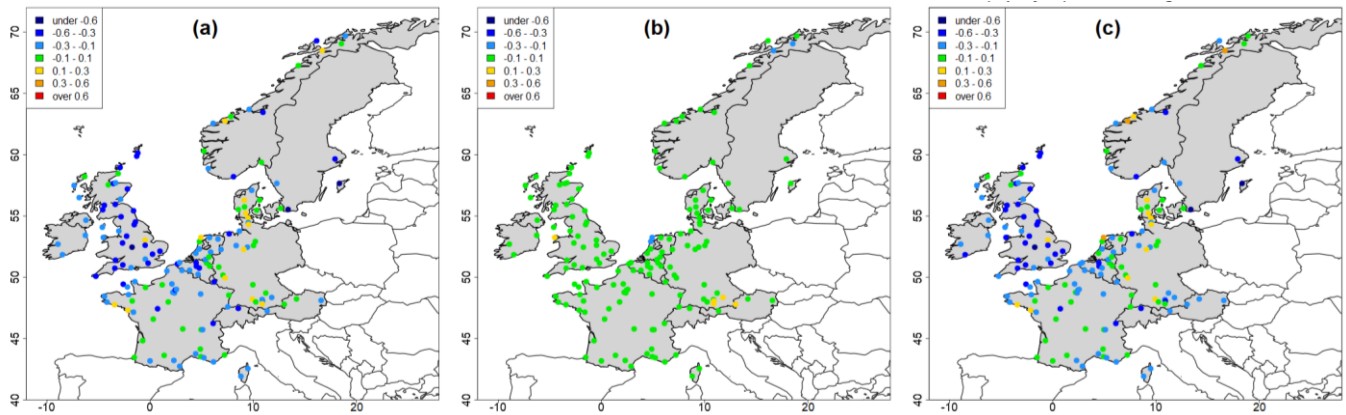

**Figure 4: the linear trend (in % per year) from 1973 to 2022, for (a) ISD gusts, (b) ERA5 winds, and (c) their difference.**


Using the same method as before, each station record was processed to obtain the annual average of the top five values of wind or gust, then re-scaled with its timeseries average. Finally, the annual means are averaged over all UK stations to produce the plotted values in Figure 5a from ISD and ERA5 datasets. On the face of it, all three timeseries appear quite similar, though closer inspection reveals how ISD gusts tend to be above the other two in the first half of the record, then mostly below them

in the second half. Figure 5b shows this behaviour more clearly: the ratio of ISD gusts to ERA5 winds (black solid line) has a downward trend indicating the relativity of ISD gusts to ERA5 winds is declining over the past four decades. In contrast, extreme gusts from ERA5 have a similar trend to their winds over this period.

Observed trends in measured gusts are not without uncertainty too, due to evolving sensors and logging systems, and changes in location and surrounding land-use (e.g. Minola et al., 2016). It is feasible that the large spatial scale of the signals in Figure

4c are explained by a systematic change across many stations. Indeed, the history of U.K. gust observing systems in Sloan and Clark (2012) does indicate a nationwide change in cup anemometers from 1997 onwards. However, side-by-side testing by Sloan and Clark (2012) indicate the newer U.K. wind sensors measure about 5% higher two-minute wind speeds. Safaei Pirooz and Flay (2018) performed a comparison of similar anemometers and found measured gusts increased by a slightly higher amount of 7 to 13%. In addition, an inspection of Figure 4a reveals an area of declining gust trends in northwest Europe which

is more consistent with a change in the storm track, rather than weather station changes across quite independent national meteorological services of U.K., Ireland, Benelux and northern France. These considerations point to the declining gust trends found across the U.K. to be meteorological in nature.

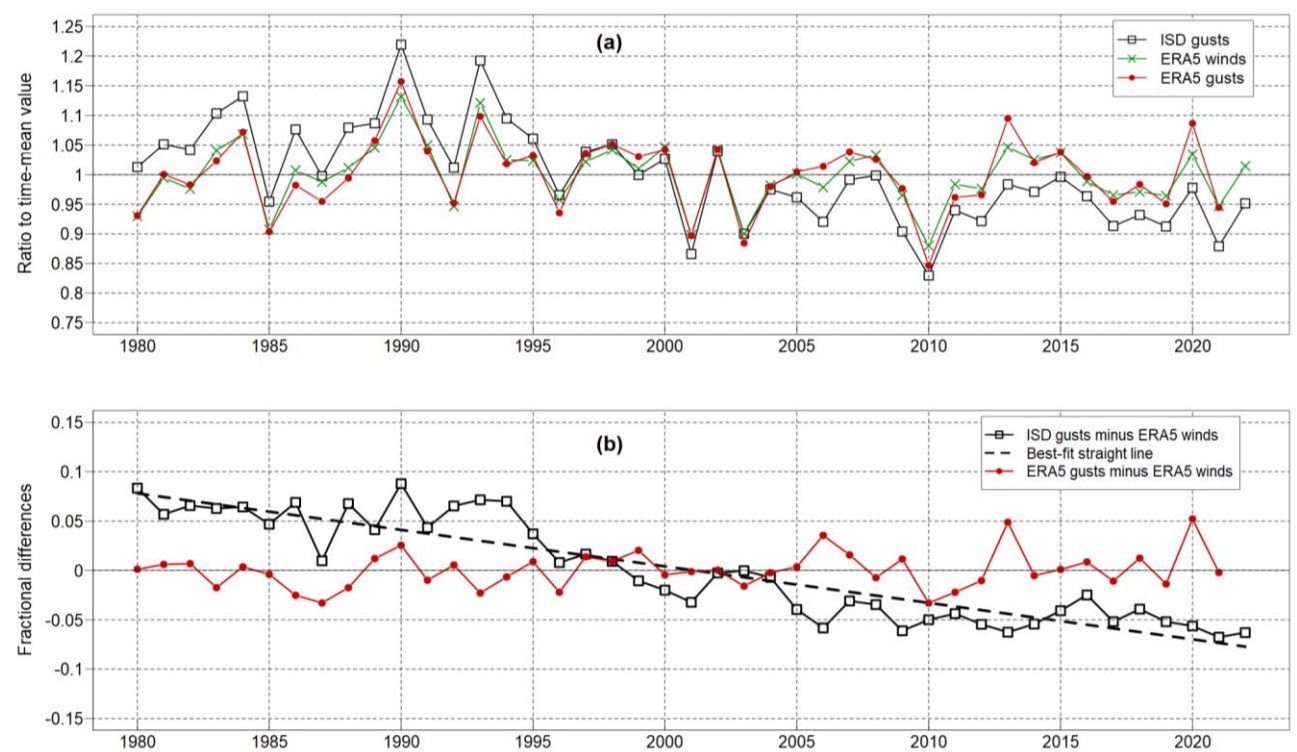

**Figure 5: (a) U.K. average timeseries for ISD gusts and ERA5 winds and gusts, re-scaled by their climate means, and (b) timeseries of the difference between each of the gust datasets and ERA5 winds.**

In conclusion, the evidence suggests modelled losses are too high in modern times because ERA5 winds do not capture the
magnitude of declining extreme gusts from the late 20th century to the present day. The next section describes a correction to ERA5 winds to include this trend, towards forming a new modelled loss dataset.

### 4.3 The final set of historical losses

The trends in observed gusts were imposed on the ERA5 event-maximum winds to produce the final loss record. The first step
was defining the trend to imprint onto reanalyses winds. This trend is defined uniquely at each ERA5 grid cell to capture the spatial variations in Figure 4c, such as smaller-sized trends in central versus northern France. If $T_{G,S}$ is the trend in observed gusts (from ISD) from 1980 to 2022 at station $s$ in units of % per year, and $T_{W,S}$ is similar for ERA5 winds, then the difference $T_{D,S}$ is given by $(T_{G,S} - T_{W,S})$. These quantities were calculated at all 221 stations, then spatially interpolated to form $T_{D,i}$ , the deficit in the ERA5 wind trend at every grid cell $i$ of the domain, using exponential weights, as follows:

$$T_{D,i} = \frac{\sum_S T_{D,S}. \exp\left(-\frac{d_{s,i}}{A}\right)}{\sum_S \exp\left(-\frac{d_{s,i}}{A}\right)}$$

where $d_{S,i}$ is the distance from station $s$ to grid cell $i$ in km, and A is a radius of influence and set to 75 km to produce reasonably smooth variations across the domain.

Figure 6 shows a map of the annual difference in trend between observed station gusts and ERA5 winds using the above equation. Negative values are dominant in a swathe from about 48°N to 60°N, where much wind damage occurs due in large

part to higher exposure density. The most negative values are in Ireland and the U.K., with quite neutral values in the northernmost and southernmost parts of the domain, off the main Atlantic storm track.

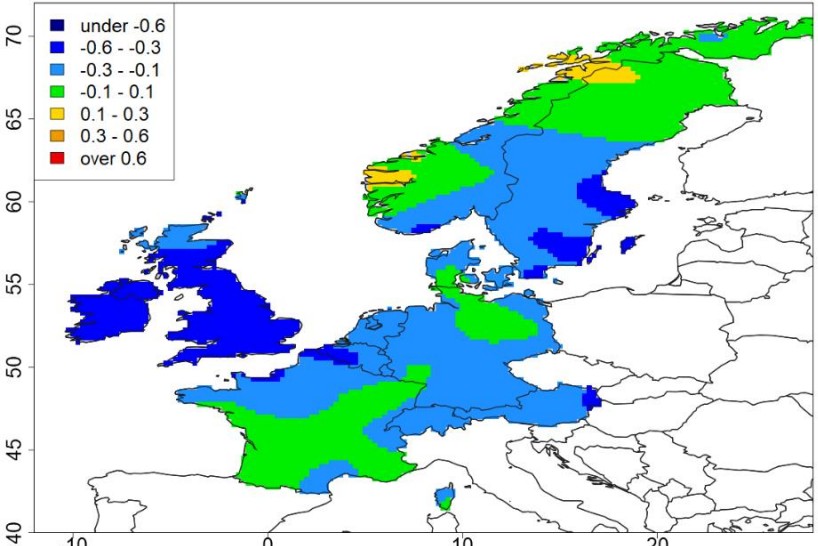

**Figure 6: map of station values of the difference in trend (% per year) between ISD gusts to ERA5 winds, over the period 1980-2022,**
**spatially interpolated to the full domain.**

A scaling factor $F_{i,y}$ is then defined for each grid cell $i$ and year $y$ from 1980 to 2022, as follows:

$$F_{i,y} = 1.0 - T_{D,i}.(y - 1980)$$

By design, scaling factors are smaller in later years, if observed gusts decline more strongly than ERA5 winds in the area.

There is little information to define scaling factors prior to 1980. The difference between ISD gusts and ERA5 were extended back to 1973 with a reduced set of stations, and it was found that the 1970s had similar or slightly lower scaling factors compared to the early 1980s. There are almost no ISD gust data before 1973, and the absence of information suggests a long-term average scaling factor is appropriate. Following these considerations, the values in 1950 to 1970 were defined to equal the long-term average scaling factor in 1980-2022, then the years 1971 to 1979 were based on a linear interpolation in time

between values in 1970 and 1980. Finally, the scaling factors defined for each year from 1950 to 2022, for each grid cell *i*, are applied to the appropriate event-maximum winds from ERA5.

Figure 7 shows the annual losses of this new dataset (blue bars) alongside the original data plotted in Figure 2 (red bars). Both versions are scaled such that 1990 losses equal €30 billion, and the new gust-corrected version has notably smaller values in the 21st century. This is because the trend in observed gusts compared to ERA5 winds over the past few decades is generally

quite negative (Figure 6c), hence domain-wide losses will be lower in the new dataset.

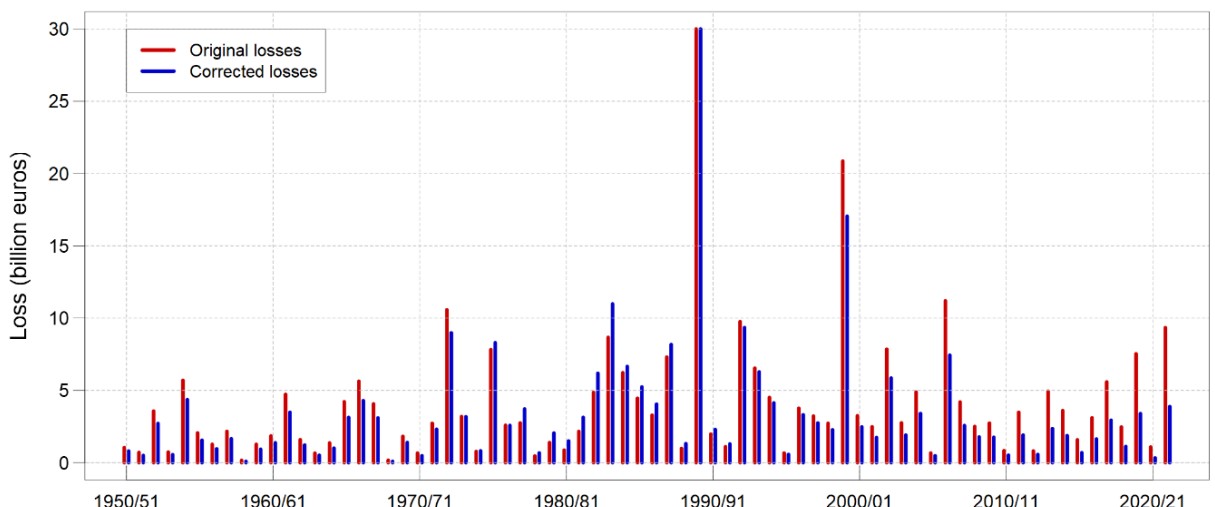

**Figure 7: timeseries of Europe-wide annual windstorm losses, both original (black bars, from Figure 2) and corrected versions.**


The modified losses lead to improvements with respect to observed losses in Table 1. Both seasons 1975/76 (incl. Capella) and 1987/88 (incl. 87J) now exceed the 2006/07 winter losses mainly caused by Kyrill. Inspection at the storm level reveals modelled losses of €8 billion for Capella and €6.5 billion for Kyrill, comparing well to observed values. While the modelled 87J loss is €5.2 billion is higher than the original value, it remains significantly below the best estimate in Table 1. The 87J

storm contained many small-scale processes such as deep convection and a sting jet which boosted its peak gusts (Browning, 2004) but may not be captured by the reanalyses winds, and further, the incurred losses were boosted by unusually severe amplification of repair costs due to demand surge (RMS, 2007) which the loss equation does not model. Overall, the relativity of these older storms to Kyrill is much better in the new dataset, though 87J has room for further improvement.

Modelled losses for extreme events in the 20th century contain a second feature of note. The losses in 1999 are lower than the

best estimate of €22 billion insured loss, and observed costs seem well defined considering the small spread in losses from different sources in Table 1. Further inspection revealed the ERA5 wind footprint for Lothar failed to capture a secondary feature which developed to the south of its main wind swathe and caused severe damage in northern Switzerland, while the

modelled Anatol loss is less than half of the value based upon PERILS, and its footprint swathe also seems less extensive (not shown). Fixing these features would raise modelled losses significantly, and much closer to the best estimate. The mere strips of wind missing from these footprints highlights the difficulty of producing accurate national losses for extreme storms.

The separate validation using PERILS losses in the 21$^{st}$ century indicates the revised loss dataset is a major improvement. Figure 8 is a copy of Figure 3 with the new, gust-corrected annual losses plotted as blue bars. The large positive bias of 107% found in the original dataset becomes a 3% deficit in the gust-corrected version. While the smallness of the bias in the new dataset is fortuitous, it was reasonable to expect a large step in the right direction from the correction.

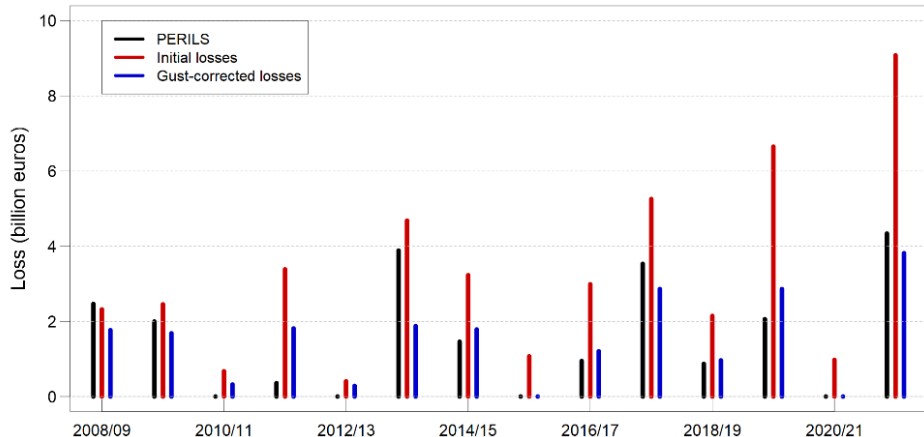

**Figure 8: a barplot of Europe-wide windstorm losses for the past 14 seasons, from PERILS and both initial and corrected modelled loss records.**

### 4.4 Discussion

There is relatively high confidence that observed windstorm losses are significantly lower this century, and that declines in observed gusts bolster this view since gusts are the hazard quantity most closely associated with windstorm damage. This leads to the question of why ERA5 winds do not carry this longer-term trend. Two different possibilities for imperfect long-term trends in ERA5 winds were mentioned earlier. First, non-stationary observing systems can drive non-meteorological trends in reanalyses (e.g. Bengtsson et al., 2004; Thorne and Vose, 2010; Wohland et al., 2019), and while Europe has had a dense observation coverage for decades, the occurrence of event peak winds is often focused on small spatial scales around fronts, and it is plausible that their representation in reanalyses has sharpened over recent decades. Future study of the intensity of fronts in European storms, preferably using observations, and the lessons in Thomas and Schultz (2019), would address this issue. A second possible cause is that the trends in gusts have quite distinct changes not present in longer time-mean winds (and the gust modelling in ERA5 does not capture either). This too is plausible, since the shorter timescales of gusts are associated with processes at smaller spatial scales, hence different trends are feasible. Mechanisms that bring upper-level winds

to the surface, such as convection (vertical and slantwise) and enhanced downward mixing near the cold front, are found to be important drivers of some of the strongest gusts in extreme storms (e.g. Browning, 2004; Fink et al., 2009). More generally, the low-level stability influences the vertical extent of mixing by small-scale mechanisms hence the gustiness of storms.

Hewson and Neu (2015) gave a very interesting example of this effect. They found that the destruction caused by storm Daria, the most damaging wind event in the U.K. for many decades, was enhanced by the weak afternoon sun in January in England and Wales: it was sufficient to reduce low-level stability which enhanced the downward mixing of momentum which in turn intensified near-surface gusts, and a little boost to strong winds can create a lot more damage. More generally, an examination of trends in low-level stability, specifically at those times when lower troposphere winds are near peak per location, may

possibly provide insights into the multidecadal reduction in storm gustiness found in the ISD observations.

A limitation of the loss dataset is also worth highlighting. The wind scaling factors in the bias-corrected dataset for 1950-79 are based on assumptions, which will need considered when interpreting the loss record in the earlier period. Research towards a better understanding of what has influenced the relativity of extreme ISD gusts to ERA5 winds over the past few decades may provide better guidance on their relativity in earlier times.

The biggest annual loss occurred in 1989/90, and was caused by the combination of a cluster of severe storms hitting dense exposure. The extent to which 1989/90 makes that late 20[th] century period appear stormy, versus the alternative – that a stormy period made 1989/90 more likely – is unclear. The new storm dataset can inform this debate. The effects of variable exposure density on annual losses are simply removed from the Klawa and Ulbrich (2003) loss equation by deleting the population term ($P_i$) to leave what is commonly referred to as a Storm Severity Index (SSI). Figure 9 shows the timeseries of standardised

losses and SSI, together with their low-pass filtered versions. The Pearson correlation of annual values of SSI and loss is 0.901, and this rises to 0.948 for the low-pass filtered versions, suggesting exposure variations play a much smaller relative role than the hazard towards temporal variations in loss. Moreover, the SSI timeseries show higher storminess levels throughout the 1980s and '90s, with 11 years above the long-term average versus three in the past 20 years. The hypothesis that the mean SSI value from 2002/03 to 2021/22 is the same as the mean from 1980/81 to 1999/00 is rejected at the 5% level, with a p-value of

0.004. A similar test on losses also rejects the hypothesis at the 5% significance level, with p-value of 0.019. The new storm dataset suggests the last two decades of the 20[th] century were stormier as a whole, which raised the probability of occurrence of extreme annual losses such as the 1989/90 season.

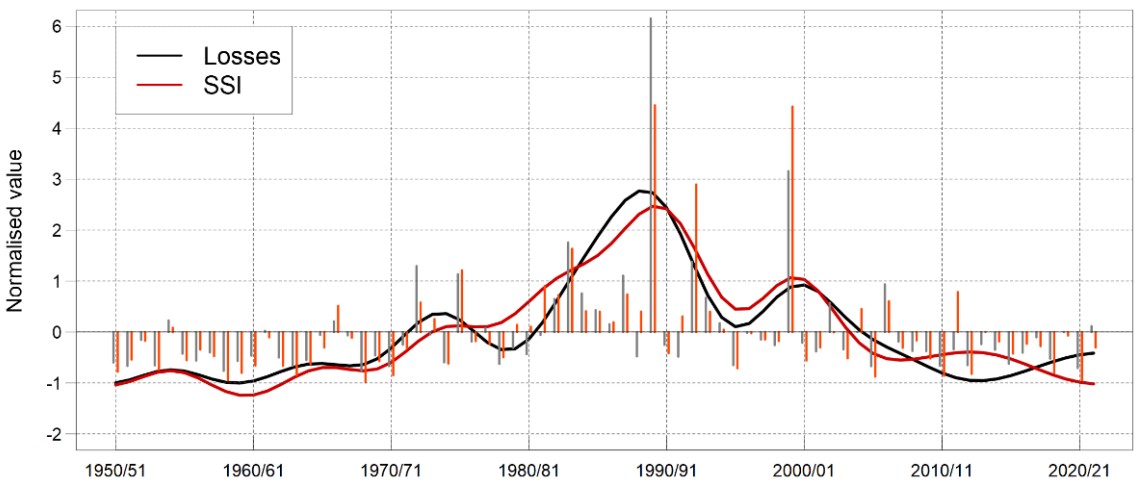

**Figure 9: timeseries of losses (black) and SSI, in normalised values, both for annual values (bars) and low-pass filtered versions with 10-yr cutoff frequency.**

## 5 Comparison of large-scale climate indices to European wind losses

The strength of connection between commonly used indices of winter-mean climate anomalies and losses is analysed in this

section, with more focus on the decadal scales that fit with the response times of the bulk of the insurance industry, though correlations at interannual timescales are presented too. All timeseries are normalised using their respective means and standard deviations for comparison purposes, as outlined in Section 2.

The NAO, Scandinavian Pattern (SCA) and the Arctic Oscillation (AO) are modes of interannual variability of circulation that are often regarded as proxies of European storminess, and Figure 10 shows their low-frequency variations alongside losses

over the past seven decades. The first finding from this comparison is how NAO values based on station data are a considerably poorer match to losses than the other three, explaining only 15% of the variance of loss at these long timescales. The main cause of this lower skill is its highest values occurring in the 2010s when losses were below average. NAO values based on fixed spatial points provide less benefit for management of windstorm risk. The second main finding is that the other three indices provide reasonably good guidance of decadal variations in losses over the whole 70 years. There is a caveat though:

the three climate indices perform relatively poorly in recent times, with the 2010s containing the second highest decadal values, in contrast to low losses. This is a concern, since the most recent period is most relevant for pricing of near-term risk, and is investigated further after presenting more results from analysis.

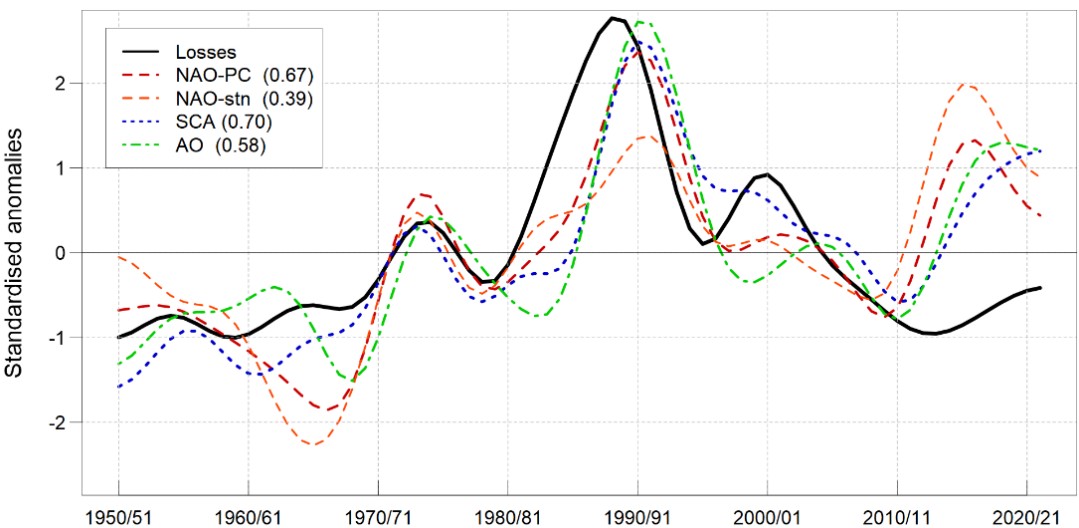

**Figure 10: low-pass filtered timeseries of European wind losses, and standard climate indices. The Pearson correlations between the climate indices and the loss dataset are shown in parentheses in the legend.**

Figure 11 provides more quantitative details on Pearson correlation values between each of the four indices and losses, at both interannual and decadal timescales. The correlations are computed both over the full time period, and a shortened version up

to 2009/10 to isolate the contribution from the recent period. Correlations are around 0.4 at interannual timescales, suggesting that even a perfect forecast of seasonal mean surface pressure anomalies is of limited benefit to pricing windstorm risk. At decadal scales, correlations rise up to 0.7, and if it is presumed that the NAO metric in Athanasiadis et al. (2020) behaves like NAO-PC and asymptotes towards 0.7, then current systems may explain around 25% of the variance in decadal losses.

Returning to the concern about the recent period, it is notable how decadal correlations of climate indices with loss fall from

about 0.85 to 0.7 when the most recent 12 seasons are included. The hypothesis that this divergence could occur randomly by sampling error was tested by calculating mean values over 2010/11 to 2021/22 for all five timeseries in Figure 10, and it revealed the values for all four climate indices were significantly different from losses at the 5% level (p-values around 0.02, except NAO-stn at 0.005). This result suggests users ought not to equate decadal variations in meteorological indices with loss anomalies in recent times. However, this evidence is merely statistical in nature and falls far short of conclusive proof that the

relationship between winter-mean winds and extreme gusts is different between the late 20[th] and early 21[st] centuries. Further research into the processes causing time-mean winds and extreme gusts would help understand whether their relationship is non-stationary, and is suggested as high priority to shore up confidence in modern-day forecasts.

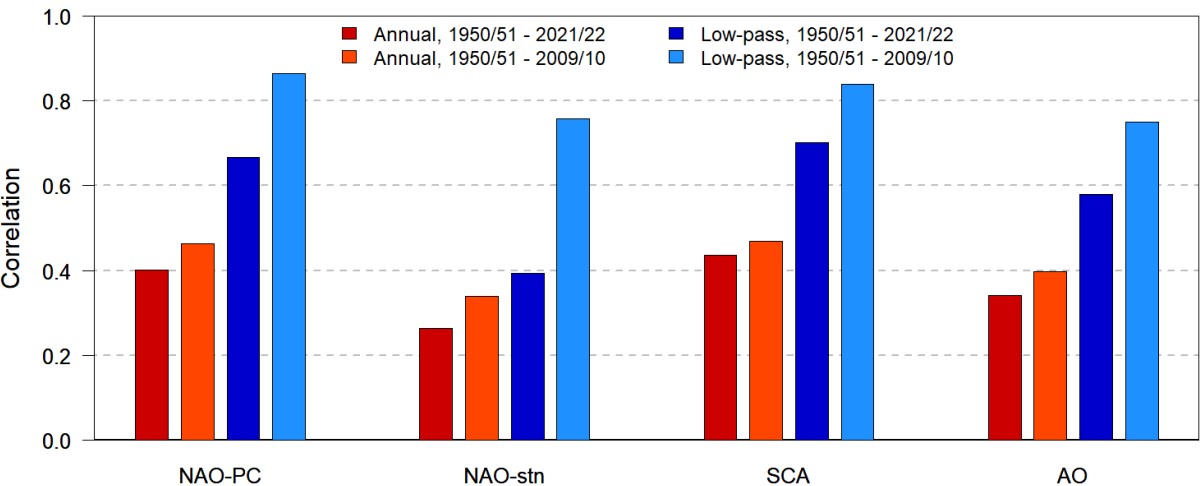

**Figure 11: correlations between each of four climate indices and losses for annual and low-pass filtered timeseries, calculated for two different time periods: the entire study period, and a shortened version excluding the final 12 seasons (i.e. 1950/51 to 2009/10).**

It is not novel to find climate indices do not explain all European storm variations. For example, Woollings et al. (2015) resolved climate variations into two timescales using a 30-yr cutoff, and their Figure 3 shows the NAO is linked to very

different changes in the North Atlantic jet between the two frequency ranges. Variations of NAO at annual-to-decadal scales were linked to latitudinal changes in the jet which is expected to have less impact on losses, consistent with the quite low correlations in Figure 11. However, similar NAO variations were associated with a strengthening and extension of the jet into Europe at longer timescales, which would be expected to have a significant impact on losses. Clearly, variations in long-term mean surface pressure patterns are not tied tightly to the occurrence of extreme event losses.

Finally, it is interesting to note how modelled losses based on uncorrected ERA5 winds have a close connection to observed climate indices in the most recent period. It suggests ERA5 extreme winds are reflecting winter-mean circulation anomalies, whereas losses have separated from time-mean winds. Though the puzzle remains as to why observed gusts and damages have a different long-term trend from time-mean winds.

## 6 Conclusions

ERA5 near-surface peak winds provide a solid foundation to build a long timeseries of European windstorm insured losses, correctly identifying those years with landmark storms and simulating the well-known multidecadal pattern of lower values in the 1960s, rising steeply to a peak in the 1980s and '90s, then decline into the 21st century. However, the recent downward trend in losses from the 1990s to the 2010s was less steep than observed. Various potential causes were considered, including the accuracy of observed losses and ERA5 winds, and a non-stationary relationship between wind and damage. It was found

that ERA5 winds simply contained a different long-term trend from damage. Further, the ERA5 winds did not match gust

trends in ISD observations, and the latter were considered reliable because they consisted of a large-scale signal measured by many coastal and inland stations. Given the close relation between damages and peak gusts, a revised loss dataset was built with observed gust trends imposed on ERA5 winds, and its decline from the 1990s to 2010s was much more consistent with observed losses. Further, it improved the estimates of event losses for some of the most severe historic storms.

The skill of ERA5 winds as the basis for storm losses is most notable given the high sensitivity of damage amounts to wind speeds. Nevertheless, its limitations should be considered too. Besides the long-term trend feature mentioned above, the footprints did not capture the extent of damage in a few key storms, such as Lothar, 87J and Anatol, and indicate the new dataset is likely to have larger relative errors in national-scale losses, compared to domain-wide. Further research could improve aspects of the final dataset. First, the long-term trend correction to ERA5 winds had no information pre-1973. A

greater understanding of the mechanisms causing the mis-match between observed gusts and ERA5 winds may help infer pre-1973 trends. Second, local storm wind details could be boosted by combining observed gust information with ERA5 winds to produce better modelled national losses. The quality of the resulting dataset would be non-homogeneous, in the sense of greater accuracy in the period of better gust data from 1970s onwards. Extra care would be needed to avoid confounding climate variability with non-meteorological trends.

The new loss dataset was used to assess some indices commonly used to summarise Europe-wide winter climate anomalies. At interannual timescales, correlations are around 0.4 and point to a modest association between climate indices and losses. At longer timescales, the indices generally have correlations around 0.7 because they capture the observed multidecadal variations of storm loss from 1950 to the early 2000s. However, all of them diverge from loss experience over the past 15 years. In a context of relatively small observational errors, there is high confidence that climate index values in the 2010s approached those last seen 30 years ago, yet a similar level of certainty that storm damages were far below those in the 1980s

and '90s. Such a decoupling was noted in previous research on the inability of the NAO to distinguish between very different changes in storm tracks over Europe. This is a key issue for insurance, because it implies the reported correlations based on the whole timeseries are not appropriate for the present-day. The available evidence suggests lower than average losses are occurring, and being driven by declining wind hazard (gusts), and standard climate indices do not reflect this reality.

Intriguingly, ERA5 extreme winds have a similar flattish trend to observed time-mean winds represented by climate indices. The non-stationary relations to the climate indices are restricted to observed extreme gusts and damages. Further investigation of the weak link between anomalies in storm damages and climate indices over the past 15 years may help connect decadal climate research to windstorm insurance.

**Acknowledgements**

The author is very grateful to two anonymous reviewers and Dr. Matthias Klawa for their advice and suggestions which considerably improved the original manuscript, and thanks the handling editor Prof. Ricardo Trigo for his valuable guidance

too. The author appreciated the many discussions of storm winds and losses with ex-colleagues at Moody's RMS, and continues to benefit from debates with new colleagues in the insurance industry.

**Code/Data availability**

ERA5 data (Bell et al. 2021) were downloaded from the Copernicus Climate Change Service (C3S) Climate Data Store at https://cds.climate.copernicus.eu/cdsapp#!/home.

The population data used in the loss index are available from https://sedac.ciesin.columbia.edu/data/collection/gpw-v4.

GSOD weather station data were downloaded from https://www.ncei.noaa.gov/data/global-summary-of-the-day/.

**Author Contribution**

The design and analysis of tests, and manuscript preparation were all performed by the author.

**Competing Interests**

The author declares no competing interest.

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
