# Peer review of "A long record of European windstorm losses, and its comparison to standard climate indices"

_Natural Hazards and Earth System Sciences, 2022_

## Author Comment (AC1)

**Referee 1**

**General comments**

*This manuscript is devoted to the analysis of the recent mismatch between the interdecadal variability of storm losses in Europe, estimated from wind data using a conventional approach, and of indices of the large-scale atmospheric circulation (teleconnections), such as NAO or AO. This lack of agreement may have critical implications for insurance companies and the general population, thereby being very pertinent and within the scope of NHESS. A new hemispheric geostrophic index (HGI), based on the 700 hPa geopotential height, is then proposed as an alternative to the more conventional indices, showing a higher correlation with the recent changes in storm losses in Europe. It is argued that HGI, being closely related to the low-tropospheric thickness rather than to near-surface conditions, explains this better correspondence. Although these findings are scientifically sounding, I found some parts a bit too speculative, thus deserving a more accurate assessment and demonstration. The text is concise and clearly written. The quality of the figures can be improved. Some revision suggestions are outlined below. Hence, I recommend the publication of this manuscript after some revisions outlined below in the specific comments.*

- The author is very grateful to the referee for their comments which have improved the quality of the manuscript.

**Specific comments:**

*Section 2.1: please describe in greater detail the datasets and the quality of the data. The average of the two reanalysis products (ERA5 and reanalysis) is also worth explaining, preferably taking into account previous research.*

- Section 2 has been expanded in the revised manuscript contains much more information about the data, in particular the historical losses because they are critical to assessment of the new dataset.
- The NCEP reanalyses are no longer used, hence averaging is not applied. The referee may wish to understand the motivation for reanalyses averaging, even though it is no longer used. It was included to support the new climate index suggested in the original manuscript. The zonal index was a function of tropical heights in the 1950s and 1960s when observational density was low hence reanalyses values are more susceptible to model biases. Based on findings in Bengtsson et al. (2004), and a recommendation in Thorne and Vose (2010) to use ensembles to reduce non-meteorological trends caused by assimilation of inhomogeneous observed data, averaging was applied. However, the revised manuscript has its focus shifted onto giving more details of loss dataset development and validation, with less material on climate indices, and no new, extended zonal index. Development of a new index with stronger connection to 21$^{st}$ century losses is now suggested for future work.

*Section 2.2: the use of 11-yr running means without values at the ends of the time series can also be improved using other more advanced methodologies, such as a low-pass filter with a cut-off frequency at 10 years.*

- The author fully agrees. The revised manuscript uses a fourth-order low-pass Butterworth filter with cut-off frequency at 10 years.

*Ln 104: Please specify "...to the present day".*

- Done (at the start of Section 3 in revised manuscript).

*Section 3.1: the limitations of the event loss equation are not stated, including their potential contribution to the recent bias. This is a very important aspect to discuss.*

- The new section 4.2 of the revised manuscript contains an expanded discussion on potential causes of the flatter trend in the original loss dataset. It considers the three main candidates: accuracy of observed trends for both loses and winds, and whether the relationship between winds and losses is stationary, as presumed by the modelling. More specifically, the third paragraph of Section 4.2 contains a discussion on whether the event loss equation contains a non-stationarity which could cause a loss trend.

*Sections 3.2 and 3.3: an assessment of the statistical significance of the trends and divergence is essential. For instance, the statements in Ln 142-143, 161-162 and 207-208 are very vague and need to be proved using robust statistical analysis of trends and inversion points.*

- Formal statistical testing of key results should have been done in the original manuscript and the author greatly appreciates this feedback.
- The revised manuscript now reports on results from statistical testing of key reported differences: between industry observed and new modelled losses, and between the timeseries of final modelled losses and climate indices in the 21$^{st}$ century. More specifically:
  - Section 4.1 contains statistical testing of the differences between the original loss dataset and the PERILS estimates.
  - Section 4.4 contains results of formal statistical testing of the difference in means between two periods: (1980/81 to 1999/00) and (2002/03 to 2021/22), for both Loss and SSI
  - Section 5 contains results from testing of the difference in means (2010/11 to 2021/22 seasons) between various climate indices and losses.

*Ln 231-232 seems to contradict the use of a zonal/hemispheric index. Please clarify.*

- Regarding the original manuscript: these lines referred to a zonal index based on surface pressures alone, and how its decadal-scale variations are strongly related to NAO and AO. In contrast, the proposed zonal index also included thickness from surface to 700 hPa, and as a result behaved differently from a zonal index based on surface pressure alone.
- Note that the proposed zonal index is removed from the revised manuscript. Instead, the study is now focused on developing a validated timeseries of windstorm losses. Development of a better climate index is suggested as a candidate for future research.

*The last paragraph of section 4.2 deserves a better discussion, including a more detailed analysis and discussion. Please revise.*

- Section 4.2 of the original manuscript concerned the observed signal-to-noise ratio of a proposed new climate index. As mentioned above, the proposal to use a new zonal index is removed from the revised manuscript, and instead suggested as a topic for future study.
- (This decision was made because the fuller description of work to establish a valid loss history has lengthened manuscript. A further extension to understand why standard climate indices are not reflecting low levels of storm damage in recent years, and to describe and analyse a new climate index, would result in a manuscript of unsuitable length.)

---

## Author Comment (AC2)

**Referee 2**

*The manuscript deals with a important topic, which is the decadal variability of windstorms affecting Europe, and the different perspectives of meteorology vs. insurance. While the idea is good, the execution has unfortunately many shortcomings, both in terms of the assumptions and analysis, and the reasoning and conclusions are very speculative on some parts. Therefore, I must suggest the rejection of the manuscript in its present form. However, I think that a strongly reworked version could be an important contribution to NHESS. Below you can find a list of the major caveat of this study, and I hope these help the author to reformulate the manuscript*

- The author thanks Referee 2 for providing very clear and informative comments in their feedback, which led to substantial improvements in the revised manuscript.
- The manuscript is thoroughly revised: descriptions of wind and loss datasets are more complete; methods to build storm footprints of wind, and their conversion to loss have been re-written and the manuscript was re-structured for clarity; analyses of results include a fuller presentation and discussion of all evidence and uncertainties. This helps clarify how conclusions are connected to the work reported in the manuscript.
- Despite all these changes, the core of the work remains the same, on developing a loss record and using it to assess the climate indices predicted by forecasting systems. The findings are similar too, and useful for both climate researchers and the insurance industry.

**Major points**

*1) datasets, line 75ff - it is not understandable to me why the AVERAGE between two different reanalysis is taken here. This will flatten the fields without necessity. Please do two separate analysis, one for each Reanalysis.*

- The NCEP reanalyses are not used in the revised manuscript, and no averaging of different reanalyses values is applied.
- For background, the referee may wish an explanation of why it was used in the original manuscript. The averaging of reanalyses was used in conjunction with the proposed new climate index. This index depended on tropical heights in the 1950s and 1960s when observational density was low hence reanalyses values are more susceptible to model biases. Based on findings in Bengtsson et al. (2004), and a recommendation in Thorne and Vose (2010) to use ensembles to reduce non-meteorological trends caused by assimilation of inhomogeneous observed data, model-averaging was applied. However, the revised manuscript has its focus shifted onto fully describing the data, methods and analysis used to make the loss dataset, and the part on climate indices is much reduced. The revised manuscript suggests the development of a new climate index, with a stronger connection to losses in the 21$^{st}$ century, as a candidate for further work.

*To look into the relationship between the large-scale patterns / pressure gradients and storms, a sub-monthly time scale would be preferable (e.g. Fink et al., 2009, doi:10.5194/nhess-9-405-2009)*

- Fink et al. (2009) did a forensic analysis of the mechanisms causing extreme gusts during storm Kyrill, and Browning (2004) performed a similar investigation of storm 87J. They both looked at small space and time scales to identify the process causing extreme gusts in different areas. These articles are held in very high regard by the author.
- However, the main aims of the manuscript – to create a timeseries of losses then measure how well the winter climate indices (used in decadal forecasts) correspond to the losses – are different from the aims of Browning, and Fink et al. Though there is one aspect of the revised manuscript which

overlaps, namely the finding that near-surface winds have a different long-term trend from gusts. The manuscript contains a possible explanation that gusts have a different mix of processes from winds, due to their different timescales. In this context, both articles (Fink et al. and Browning) are a rich source of information on gust mechanisms, and referenced in the manuscript.

*2) same section - it not clear to me why the wind gust variable from ERA5 was not used, but rather the 10m winds, as it is the former that is responsible for the damage.*

- The author agrees *observed* gust values are the better predictors of damage. However, during development, it was found that storm losses based on ERA5 modelled gusts validate poorly. ERA5 gusts are based on low-level winds with a gust parameterization added (section 3.10.4 of the IFS documentation https://www.ecmwf.int/en/elibrary/79697-ifs-documentation-cy41r2-part-iv-physical-processes), and it was found that storm damage based on ERA5 near-surface winds (no gust model scheme) validated better to observed damage totals.
- The first paragraph of Section 2.2 in the revised manuscript discusses this point.

*3) same section - The version perils data used by the author give extremely limited information and are heterogeneous in space an time, thus hampering the analysis. I would recommend to use the commercial version of the data, to which the author should have access to.*

- The author has no access to the commercial product with PERILS detailed losses.
- The publicly available data contains information appropriate for this study of Europe-wide losses. The commercial license from PERILS provides access to more refined data (by cresta, and line of business) and not used in this study.
- PERILS issue estimates of full event losses to 12 countries which together experience the majority of insured losses, and they use homogeneous data collection and methods since their inception in 2009. They are widely regarded as high-quality estimates of windstorm losses over the past 14 years. The revised manuscript includes a new description of the quality of PERILS loss estimates.

*4) data processing - this section is badly written and lacks an lot of details, and thus the methodology is not understandable. For example, I do not unterstand how "monthly data is processed into storm seasons", and the quantification of the storms and their impacts must be done at the sub-daily scale, optimally 3h for ERA5.*

- The manuscript was revised to include more information on the data and methods to make footprints of wind and loss. It also has a new structure which is intended to provide a clearer description of the data and methods. For example, the steps to create storm wind and loss footprints are gathered together into the new Section 3.
- More specifically, the revision may now be clearer on how storm winds are based on hourly data from ERA5, and damage impacts follow standard practice and estimated using event-peak winds.

*5) 3.1. basic method. The methlolody by Klawa and Ulbrich is well established and has been used by a large number of publications since. However, it is not clear for which area the data is calculated, or many other details*

- The author agrees the original version lacked full details of data and methods. The revised Section 2 includes more details on data and its processing, including a map of the studied area in Figure 1, and the new Section 3 contains more details on the method
- The aim of the revision is that a person with some experience can replicate the methods and get the same results.

*6) same section - Given that it is not clear for which area the index was calculated, and the PERILS data is only available as a single value for a subset of countries affected by a storm, the reasoning regarding Figure 1 is not understandable. A lot is section 3.2. is quite speculative. In particular, the conclusion in line 161 is not justified.*

- The first paragraph of Section 2 in the revised manuscript displays the study area of 12 countries used in this manuscript, and mentions how PERILS loss estimates cover the same 12 countries, which incur the vast majority of European windstorm losses covered by the insurance industry.
- The data and analysis leading to the conclusion in line 161 of the original manuscript is now described in much more detail, because the conclusion is key to the development of the new loss dataset. Sections 4.1 and 4.2 of the revised version contain a fuller discussion of all available evidence, and why it was concluded that reanalyses winds are the likeliest cause of a high bias in the initial loss dataset for the most recent period.

*7) section 3.3. it is not understandable for me how the loss variability of a single, small country like the Netherlands can be used to make assumptions for such a large area. Small countries have either a full hit or not hit (and thus a steeper loss curve), while larger countries like Germany or France have often partial hits, leading to a flatter curve (e.g. Karremann et al. 2014, doi:10.1088/1748-9326/9/12/124016) Thus I totally disagree with the statement in lines 170-171.*

- This comment refers to a method to adjust the long-term trend in the initial loss timeseries towards what is observed, and is replaced by a new method in the revised manuscript
- The original manuscript used a trend correction applied directly to losses, and partly based on Dutch loss history. This is now replaced by a trend correction applied to ERA5 winds, and based on observed gust trends. The difference between ERA5 and observed gust trends are described in section 4.2, while section 4.3 describes how ERA5 winds are modified to contain the observed gust trends.
- This means the statement in lines 170-171 are not relevant to the revised method.
- Though the Dutch loss history plays no part in the new trend correction, there may be a misunderstanding which is worth clarifying. The author fully agrees that the loss curve is steeper for a spatially smaller country. However, the original manuscript was referring to variations in long-term *average* losses, and the spatial scales of anomalies in *average* loss are larger than the Netherlands and contain a lot of exposure from Belgium, southeast England and North Rhine-Westphalia. Nevertheless, the use of Dutch losses to calibrate loss trends is replaced in the revised manuscript with a method regarded as better, because (a) it treats the likeliest cause of the trend bias in the initial version of the dataset (the ERA5 winds) and (b) the new calibration is based on a much bigger dataset spanning many decades and countries (GSOD weather station gusts).

*8) section 4.1 the variability of the storm activity over Central Europe is partially associated with the mentioned large-scale patters, but is best associated with a eastern shifted NAO pattern (see e.g. Fink et al., 2009). Regarding the NAO pattern I think the author is overstating the results, because a correlation of 0.60 indicates an explained variance of 36% only.*

- The manuscript assesses standard climate indices, and one of these is the Scandinavian Pattern (SCA) which represents pressure gradients over Europe, and it is found to be the climate index with closest association to storm losses, in agreement with the reviewer. (Note though that its improvement relative to NAO-PC is quite marginal.)
- The correlation values are different in the revised manuscript due to changes in filtering suggested by Referee 1, and the language used in discussion is modified too.

*9) I strongly disagree with section 4.2. - to achieve a better skill for storm impacts over Europe, a more specific index should be chosen, like a shifted NAO index, and not an even larger-scale pattern.*

- The original section 4.2 is completely removed from the revised manuscript. It was found that a detailed description of data and methods to make the initial loss timeseries, and the deeper analysis required to correct its recent multidecadal trend, followed by its comparison to standard climate indices, amounted to a significantly longer manuscript than the original.
- The reason why current climate indices fail to simulate the steepness of loss declines from 1990s to the 2010s remains an open question, and the revised manuscript includes a suggestion that this would be a good topic for further research.

*10) same section - the author is assuming a linear relationship between winds above the boundary layer and surface gusts, which is a oversimplification as the main factor is actually the turbulence in the boundary layer (see e.g. Born et al., 2012, doi:10.3402/tellusa.v64i0.17471). Thus is particularly true for when strong convective is enbedded in the cold fronts of the storms, see again Kyrill, Fink et al., 2009).*

- As discussed above in reply to point (9), the revised manuscript does not propose a new climate index, and instead recommends it for future work, and Section 4.2 of the original manuscript is removed.
- Though this topic is beyond the scope of the revised manuscript, the referee may be interested in published work describing processes that extend above the boundary layer and cause some of the strongest gusts in extreme events. For example, Fink et al. and other studies have linked the peak storm gusts at locations to deep convection reaching far above the boundary layer, and modellers consider their near-surface gusts to be produced both by vertical momentum of the downdraft being deflected at the surface, and the transport of upper-level momentum to the surface.

*11) Conclusions - given that the analysis has many weaknesses, the conclusions are unfortunately highly speculative.*

- The author appreciates the referee's very clear views on the manuscript.
- Several changes have been made in response to their comments: (i) the descriptions of wind and (especially) loss data are more complete, (ii) more information and data are included (especially on loss trending, and observed gusts), and (iii) there is a more detailed analysis of all evidence before drawing conclusions.
- As a result, the manuscript has grown in length and the proposal for a new climate index based on hemispheric geostrophic winds is removed. Instead, the development of a climate index more tightly coupled to recent low levels of loss is suggested for further work.

---

## Author Comment (AC3)

**Community comment**

*The author addresses an interesting topic in the insurance world. (Extratropical cyclone) Storm losses in Europe have been relatively low in the last two decades compared to the stormy 80s or 90s. Although standard atmospheric indices such as the NAO or AO, whose positive phase is associated with increased storm activity over the North Atlantic and over Europe in general, have tended to increase again in recent years, recorded storm losses (insured) seem to be below average. The author introduces a new class of hemispheric indices which, on the decadal time scale, provide an interesting explanation for the weak storm damage signal in Europe in recent years.*

*On first reading of the article, I found the writing style, findings and conclusions reasonable, but on second reading I got the impression that there are still some statistical weaknesses in the manuscript. In my opinion, these weaknesses should be corrected before a decision can be made about a publication.*

- The author thanks Dr. Klawa for reading the manuscript and giving his time to provide feedback on important aspects of storm climate. The author appreciates the opportunity to discuss these topics.
- The revised manuscript is improved by answering the questions raised by Dr. Klawa.
- Referee 1 made specific recommendations about statistical weaknesses and these are addressed in the revised manuscript.

*The author compares a storm loss signal with hemispheric indices on decadal scales. If we regard the loss history in Fig. 1, we can easily see the dominant storm loss of the year 1990, which was caused by a remarkable storm series within just 6 weeks (Daria, Vivian, Wiebke etc.). This single dominant year is probably the main reason why the shape of the loss curve in Fig 4 or 5 (11 year running mean) increases drastically after 1985 and drops down after 1996. I wonder, if we would discuss the findings of the author differently, if we remove the very specific year 1990, or replace the storm loss of that year by an average storm loss. I am afraid that we are discussing a random signal here: If we consider storm losses we should be aware, that the exact position of a storm footprint has an huge impact on the loss amount. If the storm does not cover the densely populated areas of Europe even a severe storm produces a small loss. Perhaps, the author could use number of events above a loss threshold instead of loss amounts.*

- The revised manuscript contains new material on how to interpret 1989/90 in the context of the European windstorm loss history
- Specifically, the third paragraph of the new Section 4.4 addresses this issue, by analysing the new loss dataset, and a new Figure 9 is included too
- In brief, the population weighting was removed from the model loss equation to form a Storm Severity Index (SSI) reflecting storm hazard strength, and the SSI timeseries indicates the period 1980-99 was much stormier than other periods, including the 21st century
- The new analysis also finds the SSI and loss timeseries are highly correlated
- Dr. Klawa has correctly anticipated that 1989/90 is less extreme in terms of SSI, than losses. However, other seasons in that period are more anomalous in terms of SSI. In particular, the profoundly stormy January 1993 in Scotland (especially northern parts with low exposure) has more anomalous SSI than loss. It was found that the average SSI in the 1980s and 1990s are as anomalous as the losses. The SSI suggests the 1980-1999 period was much stormier as a whole, which would raise the chance of a very extreme loss year like 1989/90.

*I would be more comfortable with the new HGIs if the author could show us scatterplots with HGI vs. loss (on a yearly basis, not decadal). Do these yearly HGIs perform similair compared to NAO or AO? Otherwise*

*the impression could be given that these HGIs do work only on decadal scales and that they are just a result of a random hit, which produced the correct up and downs in the graphs.*

- The proposed new index, intended to explain recent low losses, is removed from the revised manuscript.
- The original manuscript had a roughly 50:50 weight on developing a loss dataset, then comparing it to some climate indices, including the proposed HGI. The revised manuscript is now more dedicated to describing the data and methods of the loss dataset, and its validation. The development of a new climate index, specifically to explain the low level of recent losses, is suggested as potential future work in the revised manuscript.

**Specific comments**

*Ln 27: Which are the storms exceeding 20 bn USD?*

- Figure 2 of Barredo (2010) gives storm losses in USD indexed to 2008, and when trending these to 2022 using 5% p.a. (from Klawa and Ulbrich, 2003), then Capella (1976), 87J (1987), Daria (1990) and Lothar (1999) would have losses above 20 bn USD. (Note Barredo investigates economic, rather than insured losses.)
- These storm names are added in the revised manuscript.
- The trending of losses in the 21$^{st}$ century is reviewed later in the new manuscript, and a growth of 3.5% p.a. is suggested, slightly lower than the Klawa and Ulbrich (2003) value of 5%. Note that these four named storms would have economic losses above 20 bn USD if trended using 3.5% p.a. from 2008 to 2022.

*Section 3.1: Maybe I read it over, but does the loss estimation include the whole of Europe? The losses are compared to the PERILS Data. The PERILS losses cover only selected countries.*

- The domain of the new loss dataset covers the same 12 countries included in PERILS loss estimates, which together represent the vast majority of insured losses in Europe.
- The domain of study, and PERILS losses, is shown in the new Figure 1 of revised manuscript.

*Section 3.2: The author recalibrates the loss estimation for recent years, because there seems to be an overestimation of winds in the ERA5 data. Is this a known issue for ECMWF ( https://confluence.ecmwf.int/display/CKB/ERA5%3A+data+documentation#heading-Knownissues) or is there any personal communication with ECMWF?*

- This is not in the list of known issues, and the author has not discussed it with ECMWF, to date.
- More generally, the potential for non-meteorological trends in ERA5 quantities is discussed in the manuscript, in the fourth paragraph of section 4.2, and first paragraph of section 4.4.
- Further, the revised manuscript contains an investigation of how ERA5 winds have a different multidecadal trend than observed gusts from the Global Summary Of the Day (GSOD) dataset.
  - A minor point: the key issue is how ERA5 event-max winds have a flatter trend from the late 20$^{th}$ to 21$^{st}$ centuries, rather than *"overestimation of winds"*. (ERA5 winds are almost always lower than station point observations, due to the former's reduced spatiotemporal resolution.)
- This topic is also mentioned as a candidate for further work in the revised manuscript: to explore the potential to improve ERA5 gusts with information from observed gusts.

---

## Author Response (AR2)

**Referee 2**

*First of all, I would like to acknowledge the enourmous effort put by the author in revising the manuscript according to the comments by the reviewers and the editor. The manuscript has strongly improved, and the focus on specific aspects was a good choice. While I do not fully follow some of the reasoning, I do agree now that the scientific material included here that is worth publishing.*

*Still, the starting point of the reasoning is still not true, see first sentence of the abstract "Skilful predictions of European winter climate variations at interannual and longer timescales are not used by the insurance industry despite their great exposure to windstorm damage."*

*Several risk modelling companies have been using a wide range of Reanalysis and climate model simulations to enlarge their event datasets and take the multi-decadal variations in storm activity for at least ten years, and this includes RMS, Willis, Aon, MunichRe, SwissRe, Allianz, EQE, etc ... in fact, some of them have been published in this very same journal, including (but far from exaustive)*

*https://nhess.copernicus.org/articles/14/2041/2014/*
*https://nhess.copernicus.org/articles/14/2487/2014/*
*https://nhess.copernicus.org/articles/16/255/2016/*

*Also the Stucki et al 2014 and Laurila et al 2021 paper is a good examples that should be discussed in detail, and not only shortly mentioned in the introduction. It would be excellent if the author could acknowledge this fact and discuss this in the manuscript accordingly.*

- The referee has highlighted the first sentence of the Abstract, and provides examples of studies which they think contradicts the statement.
- There is a misunderstanding: the first sentence concerns the specific research topic of forecasting future storminess at decadal timescales, whereas the referee mentions studies of various aspects of windstorm behaviour, some of which include past decadal variations.
- To the author's knowledge, the first sentence of the Abstract is true and predictions of future European winter climate are not used by insurance companies in their view of risk.
- Though the referee's feedback can be viewed in a broader context. The research community have provided many actionable insights to improve how the insurance industry manages windstorm risk. The manuscript is very focused on decadal prediction, and does not mention these positive contributions.
- Based on these considerations, the author makes two changes to the revised manuscript:
  1) The opening statements in the Abstract were modified to clarify how research has helped the industry manage risk, yet this has not happened in the field of decadal forecasting. Specifically, the first sentence is expanded to three sentences.
  2) The author agrees that general storm research highlighted by the referee is worth more discussion in the article. The new third paragraph of the Introduction contains information on how researchers have helped insurers understand various windstorm topics. Sections 2 to 5 remain focused on the study of storm loss history, and decadal variability, and do not discuss general windstorm research.